# Interdependencies between Dynamic Response and Crack Growth in a 3D-Printed Acrylonitrile Butadiene Styrene (ABS) Cantilever Beam under Thermo-Mechanical Loads

**DOI:** 10.3390/polym14050982

**Published:** 2022-02-28

**Authors:** Feiyang He, Muhammad Khan, Salem Aldosari

**Affiliations:** 1School of Aerospace, Transport and Manufacturing, Cranfield University, College Road, Cranfield MK43 0AL, UK; 2Centre for Life-Cycle Engineering and Management, Cranfield University, College Road, Cranfield MK43 0AL, UK; muhammad.a.khan@cranfield.ac.uk; 3Enhanced Composite and Structures Centre, School of Aerospace, Transport, and Manufacturing, Cranfield University, Cranfield MK43 0AL, UK; s.m.aldosari@cranfield.ac.uk; 4National Center for Aviation Technology, King Abdulaziz City for Science and Technology (KACST), Riyadh 11442, Saudi Arabia

**Keywords:** fused deposition modelling, ABS, dynamic response, damage identification, displacement amplitude

## Abstract

Acrylonitrile butadiene styrene (ABS) is the most commonly used thermoplastic printing material for fused deposition modelling (FDM). FDM ABS can be used in a variety of complex working environments. Notably, the thermo-mechanical coupled loads under complex operating conditions may lead to cracking and ultimately catastrophic structural failure. Therefore, it is crucial to determine the crack depth and location before a structural fracture occurs. As these parameters affect the dynamic response of the structure, in this study, the fundamental frequency and displacement amplitude response of a cracked 3D-printed ABS cantilever beam in a thermal environment were analytically and experimentally investigated. The existing analytical model, specifically the torsional spring model used to calculate the fundamental frequency change to determine the crack depth and location was enhanced by the proposed Khan-He model. The analytical relationship between the displacement amplitude and crack was established in Khan-He model and validated for the first time for FDM ABS. The results show that a reduced crack depth and location farther from the fixed end correspond to a higher fundamental frequency and displacement amplitude. An elevated ambient temperature decreases the global elastic modulus of the cracked beam and results in a lower fundamental frequency. Moreover, a non-monotonic relationship exists between the displacement amplitude and ambient temperature. The displacement amplitude is more sensitive to the crack change than the fundamental frequency in the initial stages of crack growth.

## 1. Introduction

Fused deposited modelling (FDM) is used in 3D-printed plastic products extensively. It refers to heat thermoplastic materials, extrudes filaments from the nozzle and deposits on growing works layer-by-layer [1]. FDM is now gaining widespread attention from industry and academic research. Long et al. introduced the FDM for biomedical and pharmaceutical applications. FDM technique can create customized drug delivery devices that contain an accurate dose of medicine(s) and provide controlled drug released profiles [2]. Wong and Pfahnl used ABS to print surgical instruments such as a sponge stick, towel clamp, scalpel handle, and toothed forceps. All surgeons agreed that the printed smooth and tissue forceps would perform adequately in simulated surgical tasks [3]. Espalin et al. reported FDM device applications in the next-generation space exploration vehicle. The National Aeronautics and Space Administration (NASA) launched an FDM CubeSat Trailblazer in November 2013 to demonstrate its durability in extreme environments [4]. Some research discussed the applications of FDM ABS models for wind tunnel testing. The research results showed relatively good agreement between FDM and conventional manufactured models. The FDM model can replace metal to measure aerodynamic characteristics and verify aerodynamic data obtained in the aerospace industry, even at transonic speeds. [5,6].

Long periods of operation under these extreme conditions pose a critical challenge to the stability of the FDM structure. Often these extreme environments are accompanied by complex cyclic loads, and fatigue fracture due to cyclic loads is the most common failure in mechanical structures. To prevent fatigue fracture, predicting and estimating the severity of structural damage is particularly critical.

Non-destructive testing (NDT) techniques are commonly used to investigate the damage in mechanical structures [7]. As a representative NDT technique, the vibration method has gained considerable attention from industries and academics in recent years [1,8]. In general, a crack in a structure changes the local stiffness, thereby affecting the structural global dynamic responses such as the natural frequencies, displacement amplitudes, and mode shapes. These dynamic responses contain information regarding the locations and sizes of the crack(s). Many researchers have focused on the vibration characteristics of cracked beams [1,7,8,9,10,11,12,13,14,15,16,17,18,19,20,21,22,23,24,25,26,27,28,29,30,31,32,33,34,35,36,37,38,39,40,41,42,43,44,45]. Ostachowicz and Krawczuk (1991) modelled the crack as a torsional spring model and calculated the natural frequencies of single-sided and double-sided crack cantilever beams [46]. The analytical and experimental results highlighted that the natural frequency of the cracked beam decreased as the crack grew. Several studies have been performed to investigate the dynamic response of a cracked beam by using a spring stiffness model derived based on fracture mechanics [8,10,12,13,16,19,20,28,41,47,48,49]. However, the torsional spring model can only represent the shallow cracks. Moreover, certain researchers investigated the relationship between the natural frequency and crack propagation [8,9,10,11,12,13,14,15,16,19,25,26,34,44]. Mode shapes were applied to estimate the location and size of damage [18,22,33,43]. Other researchers proposed that changes in the curvature modes could identify the crack [15,30,50]. Zai et al. (2019) evaluated the natural frequencies and mode shapes for cracked aluminum cantilever beams [38].

Certain review articles reported a comprehensive examination of various vibration-based damage detection methods that used the frequency, mode shape, or changes in the curvature mode for assessment. However, research on the relationship between the displacement response of the structures and damage severity remains limited. Most existing research focused on cantilever beams by considering the Euler–Bernoulli beam theory or Timoshenko–Ehrenfest beam theory as the cantilever beam is a significant element in real engineering applications. Such beams can withstand high levels of mechanical loadings [23,37,51]. However, the complex external environment introduces additional thermal loads for FDM structures in addition to the mechanical loads. The coupled thermo-mechanical loads and elevated temperatures can affect the structural dynamic response. In this context, only a few researchers have considered the effect of the thermal environment on vibration analysis. Khan et al. (2015) analytically calculated and experimentally measured the fundamental frequency for a non-prismatic aluminum 1050 cantilever beam at different temperatures. The different ambient temperatures were represented by a change in the elastic modulus [31]. Furthermore, Zai et al. (2020) investigated the interdependencies of the crack depth and location on the dynamic response of the aluminum cantilever beam under thermo-mechanical loads. The analytical, numerical, and empirical results showed that an increased temperature corresponded to a reduced structural natural frequency. Several other studies reported the same conclusion [7]. Gupta et al. (2017) investigated the fundamental frequency of a cracked isotropic aluminum microplate under a thermal environment. However, the effect of the thermal environment was considered in terms of the moments and in-plane forces arising due to the temperature [39]. Gillich et al. (2019) proposed a damage detection method based on multi-modal analysis in variable temperature conditions. As a fixed–fixed steel beam was considered, the internal load caused by the increased temperature and critical load caused by buckling were introduced in the analytical model [27].

Notably, the abovementioned studies were based on metal structures, and only a few studies have been conducted to evaluate the vibration response of cracked FDM beams. Zai et al. (2019) proposed a method to in situ predict the depth of a propagating crack in an FDM ABS beam via natural frequency measurements. It was noted that FDM ABS had more residual fatigue life than a metallic structure at the same frequency drop [45]. Moreover, only a few studies have considered the effect of thermal loads on the dynamic response of FDM structures. Baqasah et al. (2019) evaluated FDM ABS beams’ natural frequency and displacement amplitude with different crack locations and sizes under thermal loads. The natural frequency response weakened at larger crack depths and temperatures as the reduction in the elastic modulus from 25 °C to 70 °C led to a reduced natural frequency. Moreover, a random amplitude behavior was observed in crack propagation tests [1].

The comprehensive review of the dynamic response of the structure indicates that the existing research [1] lacks an analytical model for the relationship between the vibration displacement response and damage severity for FDM polymers. In general, it is simpler and more convenient to measure the structural displacement response than to monitor the mode shapes in practical application scenarios. Moreover, it is of significance to develop a dynamic response model that can consider the thermo-mechanical loads for FDM polymers to reflect the actual working conditions accurately.

Considering these aspects, as an extension of the previous study [1,52], the natural frequency and displacement response for a cracked FDM ABS beam are analytically investigated. The influence of the thermal loads caused by the ambient temperature is investigated. Moreover, an optimized analytical model, Khan-He model, which can accurately calculate the natural frequency of the FDM cantilever beam with cracks, is established for the first time. By determining the orthogonality of the mode shapes, Khan-He model can be used to calculate the displacement response of the cracked beam. The analytical results are validated by tests. The Khan-He model is expected to be used as an NDT method for FDM ABS structures.

## 2. Analytical Modelling

### 2.1. Problem Description

The FDM ABS cantilever beam was chosen as the structure of interest as most prior studies [7,36,53] considered beam structures and cantilever beams as a significant element in real engineering applications [23,37]. The dynamic response of cracked beams under thermo-mechanical loads was modelled and validated.

### 2.2. Cantilever Beam Geometry

The geometry of the beam is shown in Figure 1. The thickness of the beam was 3 mm. An initial seeded crack with a width and depth of 0.2 mm and 0.5 mm, respectively, was present on the top surface, close to the fixed end of the beam to ensure the maximum stress concentration. Three crack locations were considered, at a distance of 5, 15 and 25 mm from the fixed end of the beam. An extra-lightweight accelerometer (mass: 0.6 g) from the PCB^®^ company (Depew, NY, USA) was attached to the beam’s free end.

### 2.3. Crack Size Modelling

The existence of cracks changes the local stiffness of the structure. A larger crack depth decreases the local stiffness. Consequently, the crack in the beam can be represented by a torsional spring, as shown in Figure 2 [46].

Ostachowicz and Krawczuk (1991) modelled the torsion spring stiffness corresponding to an open single-sided crack on a beam, as indicated in Equations (1) and (2).
(1)$$k=\frac{EbH^{2}}{72\pi f_{r}\left(\frac{a}{H}\right)}$$
(2)$$f_{r}\left(\frac{a}{H}\right)=0.6384{\left(\frac{a}{H}\right)}^{2}-1.035{\left(\frac{a}{H}\right)}^{3}+3.7201{\left(\frac{a}{H}\right)}^{4}-5.1773{\left(\frac{a}{H}\right)}^{5}\;+7.553{\left(\frac{a}{H}\right)}^{6}-7.3324{\left(\frac{a}{H}\right)}^{7}+2.4909{\left(\frac{a}{H}\right)}^{8}$$

Most of the existing studies [1] directly used Equations (1) and (2) as the relationship between the spring stiffness and crack depth. However, in this study, it was assumed that the stress on the crack equals the bending stress on the beam surface owing to the constant moment. Notably, such approximate stiffness equations are suitable only for cracks near the surface. Such assumptions may lead to increasing errors in the spring stiffness with larger crack propagation depths. Therefore, a novel mathematical relationship is introduced, as shown in Equation (3),
(3)$$k=\left(\frac{H-a}{H}\right)\times \frac{E_{T}bH^{2}}{72\pi f_{r}\left(\frac{a}{H}\right)}$$

The additional term $\left(H-a\right)/H$ ensures that the spring stiffness tends to zero as the crack depth approaches the thickness of the beam, consistent with the actual situation. The elastic modulus $E_{T}$ is a temperature-dependent value. The difference in the modified stiffness and originally modelled stiffness is illustrated in Figure 3.

### 2.4. Natural Frequency of the System

The cracked beam is represented by two separated full beams connected by a torsional spring, as shown in Figure 2. Similar to the existing studies, the classical Euler–Bernoulli beam theory is applied separately for the left and right beams. As there is only one fixed end on the left of the beam, the thermal expansion owing to thermal loads (elevated ambient temperature) does not introduce a force and moment. Only the fundamental frequency of the beam is calculated, and thus, the damping effect of the beam is neglected. The governing differential equation for the two beams can be expressed as Equation (4).
(4)$$E_{T}I\frac{\partial^{4}y_{L}\left(x,t\right)}{\partial x^{4}}+\rho A\frac{\partial^{2}y_{L}\left(x,t\right)}{\partial t^{2}}=0\;E_{T}I\frac{\partial^{4}y_{R}\left(x,t\right)}{\partial x^{4}}+\rho A\frac{\partial^{2}y_{R}\left(x,t\right)}{\partial t^{2}}=0\;I=\frac{bH^{3}}{12}\;\;\;A=bH$$

The effect of the thermal loads changes the value of the material elastic modulus $E_{T}$. Figure 4 shows the free body diagram of the beam section. According to this diagram, the boundary conditions of the two beams can be expressed as follows: no rotation occurs at the beam fixed end, and thus, $\partial y_{L}\left(0,t\right)/\partial x=0$; no displacement occurs at the beam fixed end, and thus, $y_{L}\left(0,t\right)=0$; deflection occurs at the spring location, and thus, $y_{L}\left(l,t\right)=y_{R}\left(l,t\right)$; the angular difference owing to spring rotation occurs, and thus, $\partial y_{R}\left(l,t\right)/\partial x-\partial y_{L}\left(l,t\right)/\partial x=\left(E_{T}I/k\right)\left[\partial^{2}y_{R}\left(l,t\right)/\partial x^{2}\right]$; the bending moment occurs at the spring location, and thus, $\partial^{2}y_{L}\left(l,t\right)/\partial x^{2}=\partial^{2}y_{R}\left(l,t\right)/\partial x^{2}$; the shearing force occurs at the spring location, and thus, $\partial^{3}y_{L}\left(l,t\right)/\partial x^{3}=\partial^{3}y_{R}\left(l,t\right)/\partial x^{3}$; no bending moment occurs at the free tip of the beam, and thus, $E_{T}I\left[\partial^{2}y_{R}\left(L,t\right)/\partial x^{2}\right]+J_{z}\left[\partial^{3}y_{R}\left(L,t\right)/\partial t^{2}\partial x\right]=0$. $J_{z}$ is neglected as it is excessively small; no shearing force occurs at the free tip of the beam, and thus, $E_{T}I\left[\partial^{3}y_{R}\left(L,t\right)/\partial x^{3}\right]-m\left[\partial^{2}y_{R}\left(L,t\right)/\partial t^{2}\right]=0$.

The general solution of Equation (4) is
(5)$$y\left(x,t\right)=Y\left(x\right)\sin\left(\omega t\right)$$

Substituting Equation (5) into Equation (4) yields Equation (6).
(6)$$\frac{d^{4}Y\left(x\right)}{dx^{4}}-\beta^{4}Y\left(x\right)=0$$
$$\beta^{4}=\frac{\omega^{2}\rho A}{E_{T}I}$$

The general solutions of Equation (6) for the left and right beams are presented as Equation (7).
(7)$$Y_{L}\left(x\right)=A_{L}\sin\left(\beta x\right)+B_{L}\cos\left(\beta x\right)+C_{L}\sinh\left(\beta x\right)+D_{L}\cosh\left(\beta x\right)\;Y_{R}\left(x\right)=A_{R}\sin\left(\beta x\right)+B_{R}\cos\left(\beta x\right)+C_{R}\sinh\left(\beta x\right)+D_{R}\cosh\left(\beta x\right)$$

The general solution indicated in Equation (7) is introduced into the boundary conditions. As at least one constant from $A_{L}$ to $D_{R}$ has non-zero solutions, according to Cramer’s rule, the characteristic Equation (8) is obtained.

The value of β is obtained by solving Equation (8). Next, Equation (6) calculates the natural frequencies of different orders for the cracked beam with an end mass.
(8)$$\left|\begin{array}{cccccccc} 1 & 0 & 1 & 0 & 0 & 0 & 0 & 0\\ 0 & 1 & 0 & 1 & 0 & 0 & 0 & 0\\ \sin\left(\beta l\right) & \cos\left(\beta l\right) & \sinh\left(\beta l\right) & \cosh\left(\beta l\right) & -\sin\left(\beta l\right) & -\cos\left(\beta l\right) & -\sinh\left(\beta l\right) & -\cosh\left(\beta l\right)\\ -\cos\left(\beta l\right) & \sin\left(\beta l\right) & -\cosh\left(\beta l\right) & -\sinh\left(\beta l\right) & \cos\left(\beta l\right)+\frac{E_{T}I}{k}\beta \sin\left(\beta l\right) & -\sin\left(\beta l\right)+\frac{E_{T}I}{k}\beta \cos\left(\beta l\right) & \cosh\left(\beta l_{L}\right)-\frac{E_{T}I}{k}\beta \sinh\left(\beta l_{L}\right) & \sinh\left(\beta l_{L}\right)-\frac{E_{T}I}{k}\beta \cosh\left(\beta l\right)\\ -\sin\left(\beta l\right) & -\cos\left(\beta l\right) & \sinh\left(\beta l\right) & \cosh\left(\beta l\right) & \sin\left(\beta l\right) & \cos\left(\beta l\right) & -\sinh\left(\beta l_{L}\right) & -\cosh\left(\beta l\right)\\ -\cos\left(\beta l\right) & \sin\left(\beta l\right) & \cosh\left(\beta l\right) & \sinh\left(\beta l\right) & \cos\left(\beta l\right) & -\sin\left(\beta l\right) & -\cosh\left(\beta l\right) & -\sinh\left(\beta l\right)\\ 0 & 0 & 0 & 0 & -\sin\left(\beta L\right) & -\cos\left(\beta L\right) & \sinh\left(\beta L\right) & \cosh\left(\beta L\right)\\ 0 & 0 & 0 & 0 & -\cos\left(\beta L\right)+\frac{m_{acc}\beta }{\rho A}\sin\left(\beta L\right) & \sin\left(\beta L\right)+\frac{m_{acc}\beta }{\rho A}\cos\left(\beta L\right) & \cosh\left(\beta L\right)+\frac{m_{acc}\beta }{\rho A}\sinh\left(\beta L\right) & \sinh\left(\beta L\right)+\frac{m_{acc}\beta }{\rho A}\cosh\left(\beta L\right) \end{array}\right|\;=0$$

### 2.5. Displacement Response of the System under Forced Vibration

A cracked beam with different crack locations and depths exhibits different displacement responses under forced resonance. This section describes the establishment of the relationship between the displacement response of the beam and crack location and depth under a sinusoidal force. Moreover, the displacement amplitude under the fundamental frequency is calculated.

#### 2.5.1. Equation of Motion

Unlike the case of free vibrations considered in the calculation of natural frequencies, as described in Section 2.4, the damping effect of the beam must be considered when the beam is subjected to forced vibration. The external damping owing to the air is neglected. The equation of motion of the beam, derived considering the internal damping stress due to the deformation along the beam and presence of the end mass and torsional spring, is presented as Equation (9).
(9)$$E_{T}I\frac{\partial^{4}y\left(x,t\right)}{\partial x^{4}}+cI\frac{\partial^{5}y\left(x,t\right)}{\partial x^{4}\partial t}+\rho A\frac{\partial^{2}y\left(x,t\right)}{\partial t^{2}}+m\frac{\partial^{2}y\left(L,t\right)}{\partial t^{2}}\delta \left(x-L\right)-\frac{\partial }{\partial x}\left[k\left(\frac{\partial y_{R}\left(l,t\right)}{\partial x}-\frac{\partial y_{L}\left(l,t\right)}{\partial x}\right)\right]\delta \left(x-l\right)\;-\frac{\partial }{\partial x}\left[cI\left(\frac{\partial^{3}y\left(l,t\right)}{\partial x^{2}\partial t}\right)\right]\delta \left(x-l\right)=0$$

The external excitation source for the forced vibrations is represented by the displacement excitation, and thus, the force on the right-hand side of Equation (9) is 0. δ represents the Dirac delta function. To simplify the calculation, the internal damping is expressed as $c=\alpha_{T}E_{T}$.

#### 2.5.2. Orthogonality of the Mode Shapes for the System

The damping effect of the beam is neglected in the calculation of the orthogonality of the free vibration mode shapes. Therefore, Equation (9) can be transformed as:(10)$$E_{T}I\frac{\partial^{4}y\left(x,t\right)}{\partial x^{4}}+\rho A\frac{\partial^{2}y\left(x,t\right)}{\partial t^{2}}+m\frac{\partial^{2}y\left(L,t\right)}{\partial t^{2}}\delta \left(x-L\right)\;+\left[k\left(\frac{\partial y_{R}\left(l,t\right)}{\partial x}-\frac{\partial y_{L}\left(l,t\right)}{\partial x}\right)\right]\delta^{\prime }\left(x-l\right)=0$$

Based on the method of separation of variables, we can assume that the solution for Equation (10) is $y\left(x,t\right)=Y\left(x\right)q\left(t\right)$, $d^{2}q\left(t\right)/dt^{2}=-\omega^{2}q\left(t\right)$. Substituting this solution into Equation (10) yields Equation (11) for mode i.
(11)$$\omega_{i}{}^{2}\left[\rho AY_{i}\left(x\right)+mY_{R,i}\left(L\right)\delta \left(x-L\right)\right]=E_{T}IY_{i}{}^{{}^{\prime }{}^{\prime }{}^{\prime }{}^{\prime }}\left(x\right)+k\left[Y_{R,i}{}^{\prime }\left(l\right)-Y_{L,i}{}^{\prime }\left(l\right)\right]\delta^{\prime }\left(x-l\right)$$

Equation (12) for mode j can be derived as
(12)$$\omega_{j}{}^{2}\left[\rho AY_{j}\left(x\right)+mY_{R,j}\left(L\right)\delta \left(x-L\right)\right]=E_{T}IY_{j}{}^{{}^{\prime }{}^{\prime }{}^{\prime }{}^{\prime }}\left(x\right)+k\left[Y_{R,j}{}^{\prime }\left(l\right)-Y_{L,j}{}^{\prime }\left(l\right)\right]\delta^{\prime }\left(x-l\right)$$

By multiplying Equations (11) and (12) by $Y_{j}\left(x\right)$ and $Y_{i}\left(x\right)$, respectively, and integrating each equation over the entire length of the beam, Equations (13) and (14) can be obtained:(13)$$\omega_{i}{}^{2}\left[\rho A\int_{0}^{l}Y_{i}\left(x\right)Y_{j}\left(x\right)dx+mY_{R,i}\left(L\right)\int_{0}^{l}Y_{j}\left(x\right)\delta \left(x-L\right)dx\right]\;=E_{T}I\int_{0}^{l}Y_{i}{}^{{}^{\prime }{}^{\prime }{}^{\prime }{}^{\prime }}\left(x\right)Y_{j}\left(x\right)dx+k\left[Y_{R,i}{}^{\prime }\left(l\right)-Y_{L,i}{}^{\prime }\left(l\right)\right]\int_{0}^{l}Y_{j}\left(x\right)\delta^{\prime }\left(x-l\right)dx$$
(14)$$\omega_{j}{}^{2}\left[\rho A\int_{0}^{l}Y_{i}\left(x\right)Y_{j}\left(x\right)dx+mY_{R,j}\left(L\right)\int_{0}^{l}Y_{i}\left(x\right)\delta \left(x-L\right)dx\right]\;=E_{T}I\int_{0}^{l}Y_{j}{}^{{}^{\prime }{}^{\prime }{}^{\prime }{}^{\prime }}\left(x\right)Y_{i}\left(x\right)dx+k\left[Y_{R,j}{}^{\prime }\left(l\right)-Y_{L,j}{}^{\prime }\left(l\right)\right]\int_{0}^{l}Y_{i}\left(x\right)\delta^{\prime }\left(x-l\right)dx$$

For two arbitrary mode shapes, $Y_{a}$ and $Y_{b}$, Equation (15) holds for any beam [54]:(15)$$\int_{0}^{L}Y_{a}{}^{{}^{\prime }{}^{\prime }{}^{\prime }{}^{\prime }}\left(x\right)Y_{b}\left(x\right)dx={\left[Y_{a}{}^{\prime \prime \prime }\left(x\right)Y_{b}\left(x\right)\right]}_{0}^{L}-{\left[Y_{a}{}^{\prime \prime }\left(x\right)Y_{b}{}^{\prime }\left(x\right)\right]}_{0}^{L}+\int_{0}^{L}Y_{a}{}^{\prime \prime }\left(x\right)Y_{b}{}^{\prime \prime }\left(x\right)dx$$

For the considered system, Equation (16) can be derived:(16)$$\int_{0}^{L}Y_{i}{}^{{}^{\prime }{}^{\prime }{}^{\prime }{}^{\prime }}\left(x\right)Y_{j}\left(x\right)dx=\frac{EI}{k}Y_{i}{}^{\prime \prime }\left(l\right)Y_{j}{}^{\prime \prime }\left(l\right)+\int_{0}^{L}Y_{i}{}^{\prime \prime }\left(x\right)Y_{j}{}^{\prime \prime }\left(x\right)dx$$

By substituting Equation (16) into Equation (13) and simplifying, Equation (17) can be obtained:(17)$$\omega_{i}{}^{2}\left[\rho A\int_{0}^{L}Y_{i}\left(x\right)Y_{j}\left(x\right)dx+mY_{i}\left(L\right)Y_{j}\left(L\right)\right]\;=\frac{{\left(E_{T}I\right)}^{2}}{k}Y_{i}{}^{\prime \prime }\left(l\right)Y_{j}{}^{\prime \prime }\left(l\right)+E_{T}I\int_{0}^{L}Y_{i}{}^{\prime \prime }\left(x\right)Y_{j}{}^{\prime \prime }\left(x\right)dx+k\left[Y_{R,i}{}^{\prime }\left(l\right)-Y_{L,i}{}^{\prime }\left(l\right)\right]\left[Y_{R,j}{}^{\prime }\left(l\right)-Y_{L,j}{}^{\prime }\left(l\right)\right]$$

By performing the same treatment for Equation (14) and subtracting the results from Equation (17), Equation (18) is obtained.
(18)$$\left(\omega_{i}{}^{2}-\omega_{j}{}^{2}\right)\left[\rho A\int_{0}^{L}Y_{i}\left(x\right)Y_{j}\left(x\right)dx+mY_{i}\left(L\right)Y_{j}\left(L\right)\right]=0$$

As the natural frequencies are distinct, $\left(\omega_{i}{}^{2}-\omega_{j}{}^{2}\right)$ does not equal zero when i≠j. Therefore, the mass orthogonality condition for the system is obtained, as shown in Equation (19):(19)$$\rho A\int_{0}^{L}Y_{i}\left(x\right)Y_{j}\left(x\right)dx+mY_{i}\left(L\right)Y_{j}\left(L\right)=0\;when\;i\neq j$$

Substituting Equation (19) into Equation (17) yields the stiffness orthogonality condition for the system, as shown in Equation (20).
(20)$$\frac{2{\left(E_{T}I\right)}^{2}}{k}Y_{i}{}^{\prime \prime }\left(l\right)Y_{j}{}^{\prime \prime }\left(l\right)+E_{T}I\int_{0}^{L}Y_{i}{}^{\prime \prime }\left(x\right)Y_{j}{}^{\prime \prime }\left(x\right)dx=0\;when\;i\neq j$$

#### 2.5.3. Displacement Response and Amplitude under the First Mode Natural Frequency

As indicated in Section 2.2, one end of the cantilever beam is fixed to the excitation source, which outputs the quasi-static sinusoidal motion. This device drives the cantilever beam in a sinusoidal displacement under the fundamental frequency. As the driving force varies with the displacement of the beam during vibration, the displacement of the shaker, as a form of external force, expressed in Equation (21) is introduced in Equation (9) [55]:(21)$$u\left(t\right)=U_{0}\sin\left(\omega_{s}t\right)$$

Therefore, the total displacement of the beam is the displacement related to the bending of the beam described in terms of the sum of the series and shaker displacement, as indicated in Equation (22).
(22)$$U\left(x,t\right)=y\left(x,t\right)+u\left(t\right)\;y\left(x,t\right)=\overset{\infty }{\underset{n=1}{\sum }}Y_{n}\left(x\right)q_{n}\left(t\right)$$

For ease of calculation, let $c=\alpha_{T}E_{T}$. Substituting Equation (22) into Equation (9) yields Equation (23).
(23)$$\overset{\infty }{\underset{n=1}{\sum }}\left[E_{T}IY_{n}{}^{{}^{\prime }{}^{\prime }{}^{\prime }{}^{\prime }}\left(x\right)q_{n}\left(t\right)\right]+\overset{\infty }{\underset{n=1}{\sum }}\left[\alpha_{T}E_{T}IY_{n}{}^{{}^{\prime }{}^{\prime }{}^{\prime }{}^{\prime }}\left(x\right)q_{n}{}^{\prime }\left(t\right)\right]+\rho A\overset{\infty }{\underset{n=1}{\sum }}\left[Y_{n}\left(x\right)q_{n}{}^{\prime \prime }\left(t\right)\right]\;+\rho Au^{\prime \prime }\left(t\right)+m\overset{\infty }{\underset{n=1}{\sum }}\left[Y_{n}\left(L\right)q_{n}{}^{\prime \prime }\left(t\right)\delta \left(x-L\right)\right]+mu^{\prime \prime }\left(t\right)\;+k\left\{\overset{\infty }{\underset{n=1}{\sum }}\left[Y_{R,n}{}^{\prime }\left(x\right)-Y_{L,n}{}^{\prime }\left(x\right)\right]q_{n}\left(t\right)\right\}\dot{\delta }\left(x-l\right)\;+\alpha_{T}k\left\{\overset{\infty }{\underset{n=1}{\sum }}\left[Y_{R,n}{}^{\prime }\left(x\right)-Y_{L,n}{}^{\prime }\left(x\right)\right]q_{n}{}^{\prime }\left(t\right)\right\}\dot{\delta }\left(x-l\right)=0$$

By multiplying Equation (23) by $Y_{i}\left(x\right)$ and integrating over the entire length of the beam, according to the orthogonality conditions of the system, as described in Section 2.5.2, Equation (24) can be obtained:(24)$$C_{1}q_{i}{}^{\prime \prime }\left(t\right)+C_{2}q_{i}{}^{\prime }\left(t\right)+C_{3}q_{i}\left(t\right)=C_{4}\omega_{s}{}^{2}\sin\left(\omega_{s}t\right)\;C_{1}=\rho A\int_{0}^{L}Y_{i}{}^{2}\left(x\right)dx+mY_{i}{}^{2}\left(L\right)\;C_{2}=\alpha_{T}\left\{\int_{0}^{L}E_{T}I{\left[Y_{i}{}^{\prime \prime }\left(x\right)\right]}^{2}dx+\frac{2{\left(E_{T}I\right)}^{2}}{k}{\left[Y_{i}{}^{\prime \prime }\left(l\right)\right]}^{2}\right\}\;C_{3}=E_{T}I\int_{0}^{L}{\left[Y_{i}{}^{\prime \prime }\left(x\right)\right]}^{2}dx+\frac{2{\left(E_{T}I\right)}^{2}}{k}{\left[Y_{i}{}^{\prime \prime }\left(l\right)\right]}^{2}\;C_{4}=\left(\rho A+m\right)U_{0}\int_{0}^{L}Y_{i}\left(x\right)dx$$

Equation (24) is a differential equation in terms of $q_{n}\left(t\right)$ and a unique homogeneous solution exists for it. This solution is calculated as the cracked cantilever beam vibrates in a steady state during the test. The solution of Equation (24) can be expressed as Equation (25):(25)$$q_{i}\left(t\right)=C_{5}\sin\left(\omega_{s}t\right)+C_{6}\cos\left(\omega_{s}t\right)$$

Substituting Equation (25) into Equation (24) yields Equation (26).
(26)$$\left[\left(\omega_{s}{}^{2}C_{1}-C_{3}\right)C_{5}+C_{2}\omega_{s}C_{6}+C_{4}\omega_{s}{}^{2}\right]\sin\left(\omega_{s}t\right)=\left[\left(\omega_{s}{}^{2}C_{1}-C_{3}\right)C_{6}-C_{2}\omega_{s}C_{5}\right]\cos\left(\omega_{s}t\right)$$

Considering the arbitrariness of time, the joint cubic equation, Equation (27), can be obtained for parameters $C_{5}$ and $C_{6}$:(27)$$\left\{\begin{array}{c} \left(C_{3}-\omega_{s}{}^{2}C_{1}\right)C_{5}-C_{2}\omega_{s}C_{6}=C_{4}\omega_{s}{}^{2}\\ C_{2}\omega_{s}C_{5}+\left(C_{3}-\omega_{s}{}^{2}C_{1}\right)C_{6}=0 \end{array}\right.$$

The solutions are presented as Equation (28).
(28)$$C_{5}=\frac{\left(C_{3}-\omega_{s}{}^{2}C_{1}\right)C_{4}\omega_{s}{}^{2}}{{\left(\omega_{s}{}^{2}C_{1}-C_{3}\right)}^{2}+{\left(C_{2}\omega_{s}\right)}^{2}}\;C_{6}=\frac{-C_{2}C_{4}\omega_{s}{}^{3}}{{\left(\omega_{s}{}^{2}C_{1}-C_{3}\right)}^{2}+{\left(C_{2}\omega_{s}\right)}^{2}}$$

By substituting Equations (25) and (28) into Equation (22), the displacement response of the cracked cantilever beam can be determined.

As the displacement amplitude of the beam is measured under the forced vibration of the fundamental frequency, the response of the higher-order modes at the first mode natural frequency is omitted, and only the first-order mode is considered. The displacement amplitude at the beam tip is expressed in Equation (29).
(29)$$U_{max}=\sqrt{Y_{i}{\left(L\right)}^{2}\left(C_{5}{}^{2}+C_{6}{}^{2}\right)+U_{0}{}^{2}}\left(i=1\right)$$

#### 2.5.4. Mathematical Approximation of the Displacement Amplitude

The analytical solution for the beam tip displacement amplitude at first-order resonance, as described in Section 2.5.3, involves certain limitations.

As the torsion spring model is used to represent the crack depth, as described in Section 2.3, the torsion spring stiffness is a fixed value for a specific crack depth. The crack is fully opened, regardless of the position and movement of the beam. However, in practice, the constantly vibrating beam subjects the crack tip to cyclic tensile and compressive loads [52]. Therefore, the actual crack depth of the beam constantly varies. The crack is fully opened when the crack tip is subjected to tensile stress for half a cycle, and at this point, the torsion spring model can represent the crack. However, the beam can be considered as an intact beam, and the crack is fully closed when the crack tip is subjected to compressive stresses for another half cycle.

This time-dependent actual effective crack depth affects the structural response of the beam. In particular, the displacement amplitude of the actual beam lies between those of an intact beam and a fully opened crack beam. However, the time-independent torsion spring model cannot represent this phenomenon. Therefore, the analytical model presented in Section 2.5.3 is bound to be inaccurate compared to the actual situation. In this study, as shown in Equation (30), a modified mathematical model is considered to reduce this error.

The term $4\left(U_{max,a=0}{}^{2}-U_{0}{}^{2}\right)\left(U_{max}{}^{2}-U_{0}{}^{2}\right)/{\left(\sqrt{U_{max,a=0}{}^{2}-U_{0}{}^{2}}+\sqrt{U_{max}{}^{2}-U_{0}{}^{2}}\right)}^{2}$ represents the displacement amplitude related to the bending of the beam. The form of the term is similar to the bilinear natural frequency of a breathing crack [13].
(30)$$U_{max,modified}=\sqrt{\frac{4\left(U_{max,a=0}{}^{2}-U_{0}{}^{2}\right)\left(U_{max}{}^{2}-U_{0}{}^{2}\right)}{{\left(\sqrt{U_{max,a=0}{}^{2}-U_{0}{}^{2}}+\sqrt{U_{max}{}^{2}-U_{0}{}^{2}}\right)}^{2}}+U_{0}{}^{2}}$$

As shown in Figure 5, assuming a displacement amplitude ranging from 2 mm (fully cracked beam) to 32 mm (intact beam) for the beam during crack propagation, the red curve shows the corrected mathematical model value. The modified displacement amplitude is always higher than the analytical result. Equation (30) allows the displacement amplitude of the intact beam to dominate the corrected result when the crack starts growth, however the value of the fully opened crack beam dominates the corrected result when the actual displacement amplitude approaches 2 mm.

## 3. Experimental Methodology

The analytical models proposed in Section 2 should be validated appropriately. Hence, a series of suitable experiments were designed with the consideration of dynamic thermo-mechanical load. The selected test parameters were determined, and the experimental setup was developed to monitor the structural integrity and dynamic response during the fatigue crack growth test. The temperature-dependent elastic modulus value in analytical models was also measured.

### 3.1. Specimen Preparation

As shown in Figure 6, the specimen, which has the geometry shown in Section 2.2, was printed using the Ultimaker^®^ 2+ printer (Utrecht, The Netherlands), with the printing parameters listed in Table 1. The other settings corresponded to the recommended or default values in the Ultimaker^®^ Cura software.

### 3.2. Experiment Scheme

The experiments involved two parts: First, the fundamental frequencies corresponding to beams with different crack locations and depths were measured, along with the displacement amplitude at the beam tip under the first mode forced vibration. Next, dynamic mechanical analysis (DMA) tests were conducted for the specimens cut from the previous beams. The elastic modulus of the beam at different temperatures was determined.

Three crack locations were considered in the experiment. The effect of thermal loading on the dynamic response was evaluated by varying the ambient temperature to 50, 60 and 70 °C. The temperature range was determined by a Differential Scanning Calorimetry (DSC) test as in previous studies [56,57,58]. A Mettler Toledo Differential Scanning Calorimeter, DSC Q 2000 (Columbus, OH, USA), was used to measure the glass transition temperature of Ultimaker red ABS. It heated the material from 25 °C to 200 °C at a rate of 20 °C/min under a nitrogen atmosphere. The test result is shown in Figure 7. It exhibits glass transition temperatures $T_{g}\approx 79.83\;{}^{\circ}C$ and $T_{g}\approx 94.49\;{}^{\circ}C$. Similarly, Braconnier et al. found two $T_{g}$ for Ultimaker white ABS shown in Figure 3 in their paper [59]. It is much lower than ABS from other sources due to a higher concentration of polybutadiene. The structural strength may become insufficient to conduct experiments owing to softening when the ambient temperature is close to $T_{g}$ [60]. Therefore, the upper limit was set as 70 °C.

Therefore, experiments were conducted in nine different configurations as each of the three crack locations was evaluated at three temperatures. Three identical specimens were manufactured for each configuration and tested in the same conditions to ensure the accuracy of the experimental results.

### 3.3. Experimental Setup and Procedure

#### 3.3.1. Dynamic Response Measurement

The experimental setup is the same as previous studies [52]. The initially seeded cracked ABS specimen was fixed on a V55 shaker manufactured by Data Physics (Hailsham, UK). The signal generator produced a sinusoidal output for the power amplifier, which transmitted the signal to the shaker. A mica band heater was used to apply a constant thermal load on the specimens throughout the experiments. The accelerometer was fixed on the beam’s free end to measure the acceleration in real-time. The acceleration and time data acquired by the accelerometer were imported into the Signal Express software via the NI 9234 DAQ card and NI 9174 DAQ chassis and recorded.

The experiment was conducted via the following steps. First, the fundamental natural frequency of the specimen was measured three times through impact tests. The mean value of the fundamental frequency was recorded. Next, the shaker introduced the sinusoidal motion with a displacement amplitude of 2 mm at the recorded fundamental frequency. The beam was driven by the shaker and vibrated at the fundamental frequency, thereby generating the initial maximum displacement amplitude of the beam. This displacement amplitude was recorded and used to examine the crack position and initial crack depth. The continuously applied forced vibration led to crack propagation, which reduced the crack area’s local stiffness and changed the complete system’s dynamic response characteristics.

The experiment was paused when the displacement amplitude displayed in real-time through the Signal Express software was significantly reduced. The system’s fundamental frequency was re-measured and re-recorded in addition to the new crack depth acquired from the Dino-Lite digital microscope. Subsequently, the test was restarted. The procedures were repeated until the final complete fracture of the beam. The experimental procedures are shown in Figure 8. Figure 9 shows a complete crack propagation process from the initial seeded crack at the location of 10 mm to the final fracture at 70 °C, as an example. Furthermore, Figure 10 shows the SEM image for the fracture surface after the FCG test, which clearly shows the air voids in FDM ABS structure.

#### 3.3.2. DMA Test Procedure

The DMA test was conducted using the Q800 device of TA Instruments (New Castle, DE, USA). The device can measure the storage and loss moduli of a sample under different temperatures through a single-clamped cantilever beam flexural test.

The test setup is shown in Figure 11. A 40-mm-long specimen was cut from the free end of the fractured beam. The part was fixed at one end in the chamber of the DMA test machine. The temperature in the chamber gradually increased from 30 °C to 70 °C at a rate of 3 °C per minute. The free end of the specimen moved at a frequency of 1 Hz and an amplitude of 10 µm. The storage and loss moduli values were automatically calculated and recorded, which varied with the temperatures.

## 4. Results and Discussion

### 4.1. Results and Analysis for the DMA Test

#### 4.1.1. Tensile Modulus of FDM ABS

The DMA of two randomly selected samples cut from the fracture beams was evaluated. The recorded data, including the storage modulus $E^{\prime }$ and loss modulus $E^{\prime \prime }$ are shown in Figure 12. The poly2 curve fitting method was used to model the correlation between the entities and temperature, with R-square values of 76.11% and 70.61%, respectively.

As illustrated in Figure 12, as thermal expansion occurs when the temperature rises, the storage modulus of FDM ABS, which represents its elastic behavior, decreases from 2104 MPa at 50 °C to 1976 MPa at 70 °C. The decrease is more rapid as the temperature increases. This trend is similar to most materials [7,31].

The loss modulus of FDM ABS, which represents its material viscosity behavior, exhibits an opposite trend with the temperature. Specifically, the loss modulus increases from 24 MPa at 50 °C to 36 MPa at 70 °C. The viscosity of the FDM ABS leads to energy dissipation caused by friction and rearrangement. A higher amount of energy is lost when the temperature is higher. The tensile modulus $E^{*}$ of FDM ABS, as a viscoelastic material under a thermal environment, is a complex physical entity, as shown in Equation (31).
(31)$$E^{*}=E^{\prime }+iE^{\prime \prime }$$

However, compared to the storage modulus, the loss modulus is two orders of magnitude smaller, and thus, the effect of viscosity can be neglected. In other words, the FDM ABS can be treated as an elastic material in the considered temperature range. The storage modulus value is equal to the tensile modulus of FDM ABS. The trend of the tensile modulus with temperature is consistent with the empirical model shown in Equation (32) [61].
(32)$$E=E_{0}-BTe^{-\frac{T_{0}}{T}}$$

#### 4.1.2. Damping of the FDM ABS Cantilever Beam

Damping reduces oscillation amplitude due to the dissipated energy to overcome friction or other resistance forms. Generally, two types of viscous damping occur for beams: resistance of the external medium (e.g., air, water) to the motion of the beam, known as external damping; and the distributed damping stresses that occur along with the height of the beam section owing to the repeated deformation of the beam fibers, known as internal damping. We ignored the effect of external damping and considered only the internal damping owing to the fiber friction of the beam in the considered conditions. As described in Section 2.5.1, the internal damping is numerically equal to the loss modulus.

Furthermore, the damping behavior of FDM ABS can be represented using the structural damping coefficient or loss factor tanδ, which equals $E^{\prime \prime }/E^{\prime }$. As the loss modulus increases with the temperature and the storage modulus exhibits the opposite trend, tanδ always increases as the temperature rises. Specifically, tanδ increases from 0.01143 at 50 °C to 0.0183 at 70 °C, as shown in Figure 13. In terms of the microstructure, the energy dissipation in the viscoelastic layers generated by their shear deformation causes internal damping. Molecular chain slippage occurs easily among the layers, and thus, a larger amount of energy is dissipated as heat by the friction when the temperature rises.

Owing to the increased damping factor at higher temperatures, the FDM ABS cantilever beam stores less kinetic and elastic potential energy under forced vibration when subjected to the same external force, resulting in a smaller displacement amplitude. This phenomenon was also observed in the experiments.

### 4.2. Dynamic Response of the Cantilever Beam–End Mass System

Crack propagation tests were conducted for beams with different crack locations and temperatures, as described in Section 3. The analytical models for the dynamic response were developed with the original and modified stiffness values. The DMA results shown in Figure 12 were considered as the elastic modulus values in the analytical model under different temperatures. The fundamental frequency of the beam is plotted in Figure 14. Figure 15 shows the analytical displacement amplitude at the beam tip during crack propagation.

Due to the fact that MATLAB R2018a reached the set maximum number of intervals when the crack depth exceeded 2 mm at the 5 mm crack location, the approximate calculated values could not attain the target accuracy. Based on the analytical results for the crack depth ranging from 0 to 2 mm, the fitted curves were plotted for the fundamental frequency when the crack depth ranged between 2 mm and 3 mm.

#### 4.2.1. Comparison of Analytical Models with Differently Modelled Torsional Spring Stiffness Values

##### 4.2.1.1. Comparison and Analysis of Differences in the Fundamental Frequency

Figure 14 shows the experimental data and fundamental frequency calculation results of the two analytical models during crack propagation at different crack locations under different temperatures. The experimental data and calculations of the two analytical models highlight a continuous decrease in the fundamental frequency with the initial crack propagation until specimen fracture. This result is similar to that reported previously [1].

However, the comparison of the results of the fundamental frequency calculations based on the two analytical models indicates that when the crack depth approaches the beam top surface (less than one-third of the total thickness of the beam), the fundamental frequency calculated using both models is identical. Notably, as the crack depth continues to increase, the fundamental frequency curve for the original spring stiffness model tends to level off, whereas that for the modified spring stiffness model decreases rapidly and tends to decrease more rapidly with the increase in the crack depth. For example, in the case of the crack growth at an ambient temperature of 50 °C and crack location of 5 mm, the difference in the fundamental frequency obtained using the two models changes from 0.00036 Hz at a 0 mm crack depth to 0.44 Hz at a 1 mm crack depth and finally to 14.61 Hz for a 2.8 mm crack depth. In terms of the percentage relative to the original stiffness model, the difference between the two models increases rapidly from 0.001% to 84% as the crack depth increases from 0.1 mm to 2.8 mm, corresponding to nearly five orders of magnitude.

This difference in the fundamental frequency as the crack growth can be attributed to the use of the spring model of the crack size, as described in Section 2.3. In the original modelling process of the spring stiffness, it was assumed that the stresses at the surface of the bending beam crack location and stress field at the crack tip were of the same magnitude. This estimation is applicable only for shallow cracks close to the surface of the beam. As the crack depth increases, the error in the stresses increases larger. Even when the beam is completely fractured, the original spring stiffness model still estimates a non-zero value, whereas, in reality, the spring stiffness model must yield a value of zero. In the proposed Khan-He model, the additional term $\left(H-a\right)/H$ is introduced, which ensures that the stiffness coefficient of the spring tends to zero when the beam is completely fractured. Therefore, the fundamental frequency of the fractured beam is 0 Hz, consistent with the actual scenario.

The comparison of the results of the analytical models with the experimental data indicates a significant difference in the accuracy of the predicted values of the two models, especially at large crack depths. As shown in Figure 14, the experimental data are further distributed along the curves of the fundamental frequency value obtained using the Khan-He model.

To examine the difference quantitatively, Figure 16 shows the relative differences of the two spring stiffness models against the experimental data. In terms of the fundamental frequencies for different crack locations at different temperatures, the overall difference between the values obtained using Ostachowicz model and experimental data is considerably greater than pertaining to the Khan-He model.

Moreover, the difference in the fundamental frequency with crack growth is similar in the conditions corresponding to each subplot. An example of crack propagation under 50 °C at the 5 mm crack location is considered. Table 2 summarizes the experimental data, corresponding analytical model calculation results, and differences associated with the crack depth measured during the crack propagation test. Figure 17 shows the relative difference between the results obtained using the two analytical models and experimental results. Table 2 and Figure 17 highlight the increasing difference between the fundamental frequencies calculated using Ostachowicz model and the experimental data during the crack propagation. The difference reaches nearly 40% when the crack depth is 2.396 mm.

In contrast, the fundamental frequencies determined using Khan-He model are in agreement with the experimental data, with a difference of less than 5% when the crack propagates from the initial depth of 0.317 mm to 2.026 mm. The difference in the fundamental frequency corresponding to crack depths of 0.963 mm and 1.279 mm is only 0.08% and 0.07%, respectively. Note that to determine the analytical model fundamental frequency at a crack depth of 2.396 mm, a second-order polynomial fit was used as the accuracy yielded by the MATLAB approach was inadequate; the use of this approach likely led to the slightly higher discrepancy (20.55%) between the value and the experimental data.

Overall, Khan-He model yielded fundamental frequency values that were similar to the experimental data. Compared to Ostachowicz model, accurate estimates of the fundamental frequency response of the beam could be obtained for specific crack locations and depths. In the following discussion of the frequency response of the cracked beam, only the fundamental frequency determined using Khan-He model is considered.

##### 4.2.1.2. Comparison and Analysis of the Differences in the Displacement Amplitude

Figure 15 shows the analytical and experimental relationships between the beam tip displacement amplitude and crack depth at first-order resonance. As in the case of Figure 14, both the analytical model results and experimental data indicate that the displacement amplitude decreases as the crack grows.

The difference in the results obtained using the two analytical models is not significant when the crack is close to the surface of the beam. However, as the crack propagates, the difference between the two values becomes progressively larger, and the displacement amplitude determined using Khan-He model decreases more rapidly than that obtained using Ostachowicz model. Eventually, the value of Khan-He model is significantly smaller than the displacement amplitude of Ostachowicz model when the beam is completely fractured. For similar reasons, the results are similar to those of the fundamental frequency during crack propagation. The additional term $\left(H-a\right)/H$ in the proposed model ensures that the local stiffness at the crack location is smaller than that pertaining to Ostachowicz model. Therefore, the beam oscillates similarly to the shaker vibration without displacement due to resonated bending when the beam approaches fracture. Therefore, the displacement amplitude is similar to 2 mm, as in the case of the shaker, when the crack depth increases to 3 mm. Consequently, Khan-He model with the modified torsional spring stiffness is superior to Ostachowicz model.

However, the proposed Khan-He model results are significantly different compared to the experimental data. The experimental data are consistent with the theory described in Section 2.5.4, as discussed in the section. The torsion spring model is not an ideal representation of the actual effective crack depth, which varies with time during the same cycle. Therefore, a new mathematical improved Khan-He model was proposed in Section 2.5.4. Figure 18 shows the displacement amplitudes obtained using the improved mathematical method and experiments at different crack depths.

The displacement amplitude, obtained using Khan-He and secondary improved Khan-He models, is similar as the crack approaches the beam surface. The difference between the two models gradually increases as the crack depth approaches the half beam thickness. Furthermore, the displacement amplitudes obtained using both models gradually converge to the same value, which equals the shaker’s vibration amplitude of 2 mm, when the beam is about to fracture. This difference can be attributed to the form of Equation (30). The nature of this equation causes the displacement amplitude of the improved Khan-He model to be greater than that of Khan-He model when the crack depth approaches the middle of the beam.

Figure 19 shows the average relative difference between Khan-He (the model in Section 2.5.3) and secondary improved Khan-He (the model in Section 2.5.4) models compared to the experimental data for different combinations of the crack location and temperatures. The difference between the results of the secondary improved model and experiment is smaller than that for Khan-He model. Figure 20 illustrates the difference between the results obtained using the two models and the measured values corresponding to the crack depth during crack propagation tests. The improved Khan-He model calculation is closer to the actual experimental values for most crack depths at different crack locations under different temperatures.

However, both Figure 18 and Figure 20 show that the experimental values are higher than the displacement amplitude obtained using the improved Khan-He model at specific crack depths, especially when the crack depth is between half and the total thickness of the beam (for instance, in conditions of 15 mm 50 °C and 5 mm 60 °C). This observation can be attributed to the crack closure phenomenon. To ensure that the displacement amplitude of the shaker output to the fixed end of the beam is a constant value of 2 mm during the experiment, the load amplitude on the beam varies with crack depth. The load applied to the beam to make it resonate decreases as the crack grows. The reduced load reduces the stress amplitude on the crack location, resulting in a lower stress ratio. This low-stress ratio leads to larger crack closure during beam vibration. Although the vibrated cracked beam can be assumed as a superposition of the intact and fully open cracked beam states, a larger crack closure renders the structural properties of the cracked beam to be more biased towards the intact beam. Therefore, expectedly, the displacement amplitude response of the beam tends to increase owing to the larger influence of the intact beam.

Nevertheless, the difference in the fundamental frequency obtained using the analytical model and the experimental data is considerably smaller than that for the displacement amplitude, as shown in Figure 16 and Figure 20. This phenomenon can be attributed to two aspects. First, different experimental methods were used to measure the fundamental frequency and displacement amplitude. The beam was at rest when the fundamental frequency was measured. The impact test only applied a slight disturbance to the beam. However, the beam was required to be in resonance with a 2 mm excitation to measure the displacement amplitude. This considerable resonance likely affected the crack growth and changed the displacement amplitude. Moreover, the fluctuating displacement amplitudes during beam vibration increased measurement error. Second, the heating band was unlike an environmental chamber and was relatively exposed. The vibration of the beam caused air to flow, leading to fluctuations in the ambient temperature of the specimen, thereby affecting the displacement amplitude.

#### 4.2.2. Fundamental Frequency for the Cracked Beam

##### 4.2.2.1. Influence of the Crack Depth on Natural Frequencies

The variation in the beam fundamental frequency during crack propagation is shown in Figure 14. As the two analytical models and experimental data yield similar trends for the change in the fundamental frequency with the crack depth at any crack location or ambient temperature, the relationship between the crack growth and change in fundamental frequency at an ambient temperature of 50 °C at a 5 mm crack location is considered for analysis. Figure 21 illustrates the fundamental frequency and drop percentage change during crack growth, as determined using Khan-He model and experimental data. As described in Section 4.2.1.1, the experimental data and analytical model indicate the reduction in the fundamental frequencies as the crack grows. Compared to an intact 3-mm-thick beam, the fundamental frequency at a crack depth of 2.8 mm drops from approximately 24.5 Hz to 2.5 Hz, which is only approximately 10% of the initial fundamental frequency. The beam’s cross-sectional area at the crack location decreases and the local stiffness decreases as the crack length increases. As reflected in the analytical model, the local reduction in the stiffness due to crack growth reduces the stiffness matrix, although the beam mass does not change; therefore, the calculation yields a gradual reduction in the fundamental frequency.

Notably, the change in the fundamental frequency is insignificant when the crack is close to the surface. However, as the beam approaches fracture, the change in the fundamental frequency becomes more rapid as the crack grows. The frequency decreases only by approximately 0.3 Hz (1% relative change) when a crack of 0.5 mm (one-sixth of the beam thickness) is present in the intact beam; in contrast, the frequency decreases by approximately 6 Hz (25% relative change) from the 2.5 mm crack depth to fracture. The initial monitoring of small cracks is crucial in actual working conditions. In other words, highly sensitive sensors are required to predict the crack depth when using the fundamental frequency.

##### 4.2.2.2. Influence of the Crack Location on the Fundamental Frequency

Figure 22 shows the change in the fundamental frequency during crack propagation for a beam with different crack locations. A crack located farther from the fixed end of the beam corresponds to a higher fundamental frequency of the beam. Moreover, the difference in the fundamental frequencies increases. However, as the crack depth approaches the thickness of the specimen (from approximately 2.3 mm, as shown in Figure 22), the difference between the fundamental frequencies of the beams with different crack locations rapidly converges to zero Hz. The effect of the crack location on the fundamental frequency is not significant. The maximum difference in the fundamental frequencies for the 15 mm and 25 mm crack locations is only approximately 0.95 Hz.

##### 4.2.2.3. Influence of the Temperature on the Fundamental Frequency

Figure 23 shows the fundamental frequency of the cracked beam at different temperatures. The fundamental frequency of a specimen with a fixed crack depth and location decreases gradually as the temperature increases. However, the difference in the fundamental frequency at different temperatures decreases as the crack depth increases. Similar to the crack location, the effect of the temperature on the fundamental frequency is not significant. Regardless of the crack location, the fundamental frequency of specimens with a 0.1 mm crack depth differs only by 0.8 Hz in the experimental temperature range. This difference decreases further with crack propagation to a theoretical value of zero Hz at the final fracture state.

#### 4.2.3. Displacement Amplitude for the Cracked Beam under Resonance

##### 4.2.3.1. Influence of the Crack Depth on the Displacement Amplitude

As the trends of the displacement amplitude of the specimens at different crack locations during crack propagation are similar, as shown in Figure 18, the displacement amplitude of the beam with a 15 mm crack location at 60 °C is discussed.

Figure 24 shows the change in the displacement amplitude with crack growth. The displacement amplitude decreases from approximately 36 mm for the initial intact specimen to 5 mm at the final near-fracture state. The change in the displacement amplitude is smooth as the crack depth approaches the beam top surface or when the beam is close to fracture. However, the displacement amplitude decreases rapidly when the crack depth approaches half the thickness of the beam.

The decrease in the displacement amplitude of the beam with the crack growth can be attributed to two reasons: First, the load to maintain the 2 mm displacement amplitude excitation gradually decreases as the crack propagates. In other words, the force acting on the beam gradually decreases. The decreased excitation results in smaller displacement amplitudes; Second, the increased crack depth increases the damping of the beam [62]. Consequently, the energy dissipated by damping increases during vibration, and the displacement amplitude decreases.

##### 4.2.3.2. Influence of the Crack Location on the Displacement Amplitude

Figure 25 shows the change in the amplitude with crack growth for beams with different crack locations. Similar to the frequency trend shown in Section 4.2.2.2, a crack located farther from the fixed end of the beam corresponds to a smaller displacement amplitude. The displacement amplitude of the beam is the same for different crack locations when the crack is close to the surface of the beam and propagation is initiated. Moreover, the displacement amplitudes tend to be 2 mm when the beam fractures. However, unlike the fundamental frequency change, the displacement amplitude difference corresponding to different crack locations increases as the crack depth closes to half the beam thickness (approximately 1.4 mm).

##### 4.2.3.3. Influence of the Temperature on the Displacement Amplitude

Figure 26 shows the effect of the temperature on the displacement amplitude of the cracked beam. Unlike all previous response trends, the effect of different temperatures on the displacement amplitude of cracked beams is extremely complex. As shown in Figure 26, the beam has the smallest displacement amplitude from the intact to fractured state at 50 °C. The intact beam exhibits the largest displacement amplitude at 60 °C. However, when the crack depth reaches and exceeds approximately 0.8 mm, the beam at 70 °C exhibits the largest displacement amplitude.

This seemingly random phenomenon validates the experimental results of the existing research [1]. Specifically, this phenomenon can be attributed to the combined effect of the varied elastic modulus and excitation loads under various temperatures. The overall flexibility of the beam decreases, and the displacement amplitude naturally increases when the beam is subjected to the same external excitation when the temperature rises. However, the fundamental frequency pertaining to the rise in temperature decreases. To maintain the two mm amplitude resonance, both the amplitude and frequency of the external forces loaded on the beam decrease, thereby reducing the displacement amplitude of the beam. These two opposing influences cause the final displacement amplitude response of the beam to lose its monotonicity and become more complex.

#### 4.2.4. Sensitivity of the Dynamic Response

Section 4.2.2 and Section 4.2.3 discussed the response characteristics of the fundamental frequency and displacement amplitude. This section describes the characteristics and sensitivity of the dynamic response to the crack depth and location. Notably, the response of the displacement amplitude to different temperatures, as described in Section 4.2.3.3, is too complex to be discussed.

##### 4.2.4.1. Sensitivity of the Dynamic Response to the Crack Depth

Figure 27 shows the fundamental frequency and displacement amplitude change at the beam tip for a beam with a 15 mm crack location at 50 °C. Although the fundamental frequency and displacement amplitude decrease with crack growth, their trends and differences from the initial values are different.

Figure 27b shows that the fundamental frequency of the beam changes by only approximately one Hz, although the displacement amplitude of the beam decreases by eight mm when the crack depth of the beam increases from 0.1 mm to 1 mm. In other words, a significant change occurs in the displacement amplitude of the beam, whereas the change in the fundamental frequency is extremely small at the beginning of the crack growth when the crack is close to the beam surface. Figure 27c shows that compared to the initial value, the relative difference in the displacement amplitude is always higher than that in the fundamental frequency for the same crack depth throughout the process from an intact state to the final fracture.

Therefore, the displacement amplitude is more sensitive to changes in the crack depth than the fundamental frequency response of the fractured beam in the early stages of crack growth. In reality, the detection of small initial cracks is extremely critical. The accurate prediction of initial cracks can often prevent the catastrophic failure of the structure even if the cracks continue to grow. This finding suggests that using the displacement amplitude to estimate the crack depth yields more accurate results than those obtained using the fundamental frequency for small cracks. In other words, the accuracy requirements for the sensor can be lowered if the displacement amplitude is measured. The fundamental frequency and displacement amplitude can be analyzed in combination for longer cracks to determine the crack depth accurately.

##### 4.2.4.2. Sensitivity of the Dynamic Response to the Crack Location

Figure 28 shows the fundamental frequency and displacement amplitude change for beams with 15 mm and 25 mm crack locations. The crack location moves away from the fixed end of the beam. The fundamental frequency and displacement amplitude of the beam decrease.

Figure 28b shows the difference in this decreasing trend. The displacement amplitude is the most sensitive to the crack location when the crack depth is close to approximately 1.4 mm. The change in the fundamental frequency is greatest when the crack position changes from 15 to 25 mm at a crack depth of approximately 2.3 mm.

Figure 28c illustrates the relative difference in the fundamental frequency and displacement amplitude. The relative differences of the two dynamic responses for different crack locations are minor in the early stages of crack growth. However, a 1% difference in the displacement amplitude is considerably more significant than a 0.17% difference in the fundamental frequency for a 0.5 mm crack depth when the crack location changes from 15 mm to 25 mm. Furthermore, throughout the crack growth, the slopes of the displacement amplitude curve are greater than those of the fundamental frequency when the crack depth is less than 2.3 mm.

In other words, similar to the crack depth, the displacement amplitude is more sensitive to the crack location than the fundamental frequency at the critical stage of early crack growth and for most crack depths. Therefore, using the difference in the displacement amplitude to determine the crack location may yield superior results than those obtained using the fundamental frequency.

## 5. Conclusions

This paper proposes an analytical model (Khan-He model) to determine the dynamic response of a cracked 3D-printed ABS cantilever beam subjected to a thermo-mechanical load. The fundamental frequency can be modelled more precisely compared to the existing model (Ostachowicz model). The corresponding displacement amplitudes were calculated considering the crack breathing phenomenon. The experimental results validated the proposed Khan-He model.

The fundamental frequency and displacement amplitude decrease as the crack grows in terms of the dynamic responses to structural cracks. A crack located farther from the fixed end of the beam corresponds to a higher fundamental frequency and displacement amplitude. Moreover, the increased temperature reduces the fundamental frequency of the fractured beam.

The displacement amplitude at the beam tip is more sensitive than the fundamental frequency to the crack depth and change in the location at the beginning of crack growth.

## Figures and Tables

**Figure 1 polymers-14-00982-f001:**
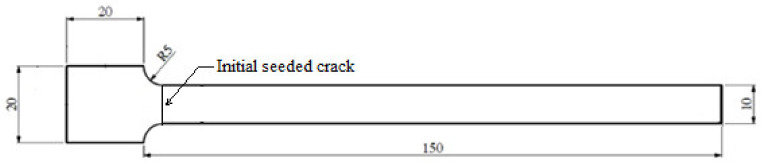
Beam geometry, with a length, width, and thickness of 150 mm, 10 mm, and 3 mm, respectively.

**Figure 2 polymers-14-00982-f002:**
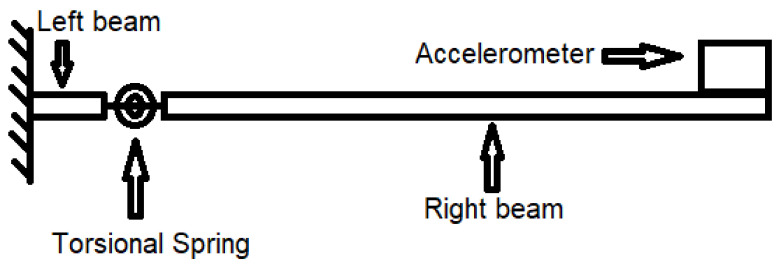
Cantilever beams–torsional spring–accelerometer system.

**Figure 3 polymers-14-00982-f003:**
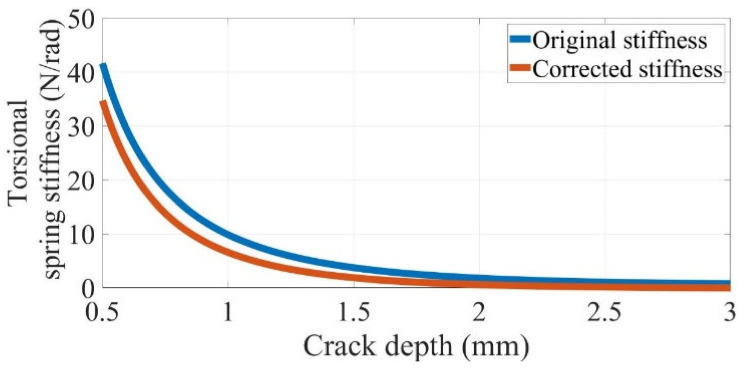
Change in the torsional spring stiffness with crack propagation in an FDM ABS beam with a thickness, width, and elastic modulus of 3 mm, 10 mm, and 1600 MPa, respectively.

**Figure 4 polymers-14-00982-f004:**
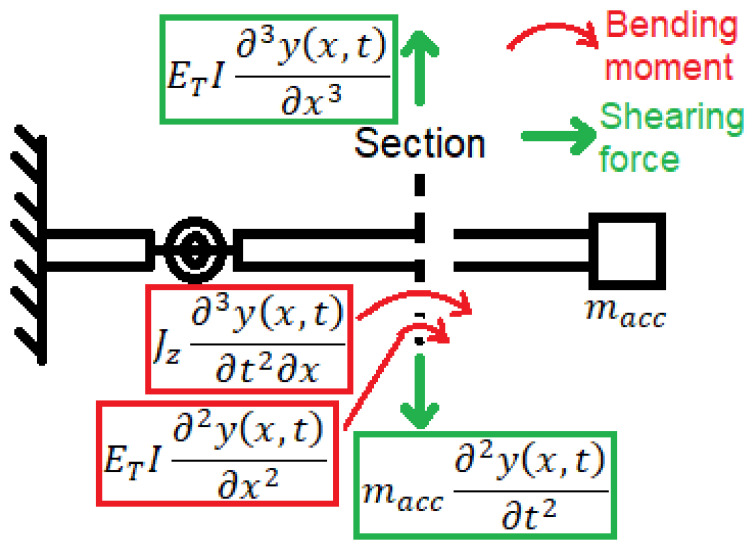
Free body diagram of a beam cross-section.

**Figure 5 polymers-14-00982-f005:**
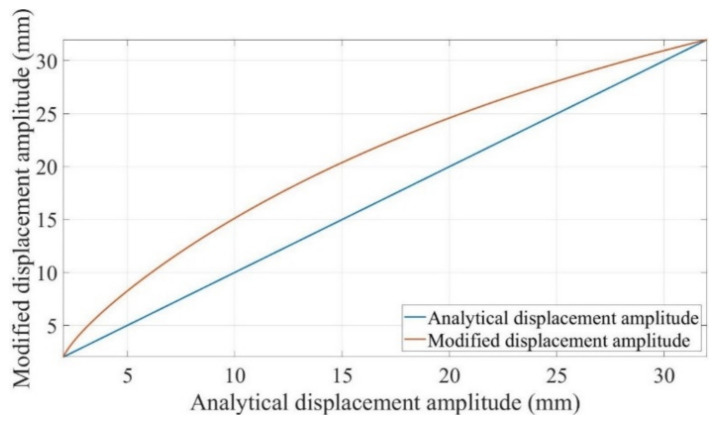
Analytical displacement amplitude and corresponding modified mathematical displacement amplitude for a range of 2 mm to 32 mm.

**Figure 6 polymers-14-00982-f006:**
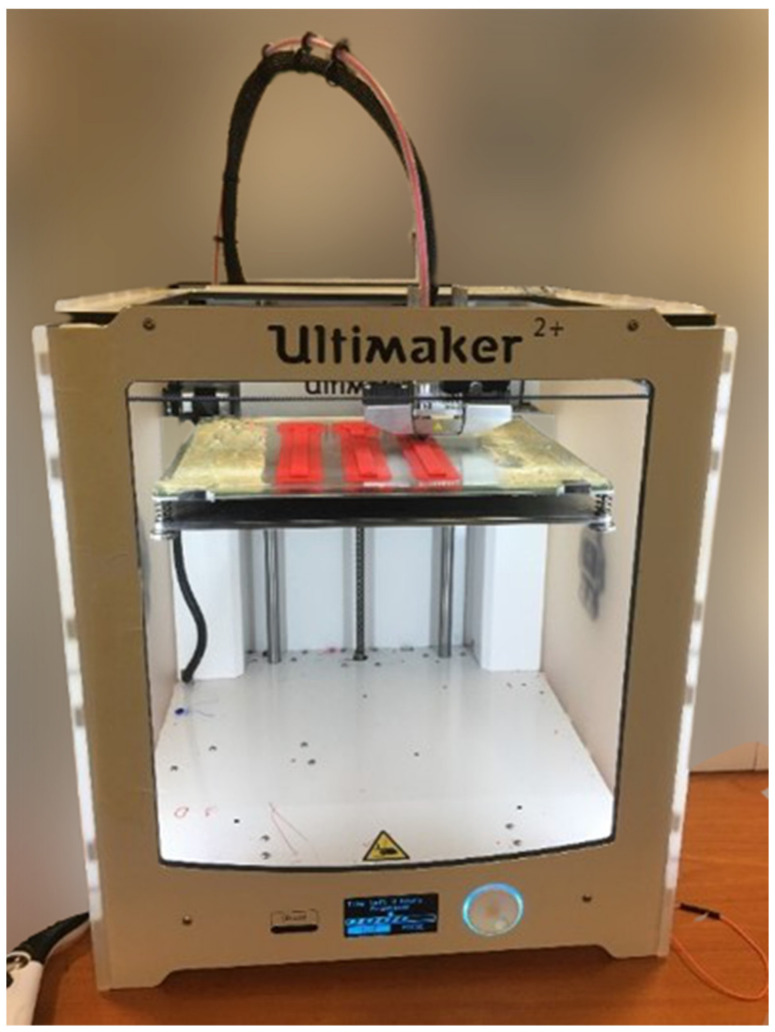
Three specimens printed using the Ultimaker 2+ printer.

**Figure 7 polymers-14-00982-f007:**
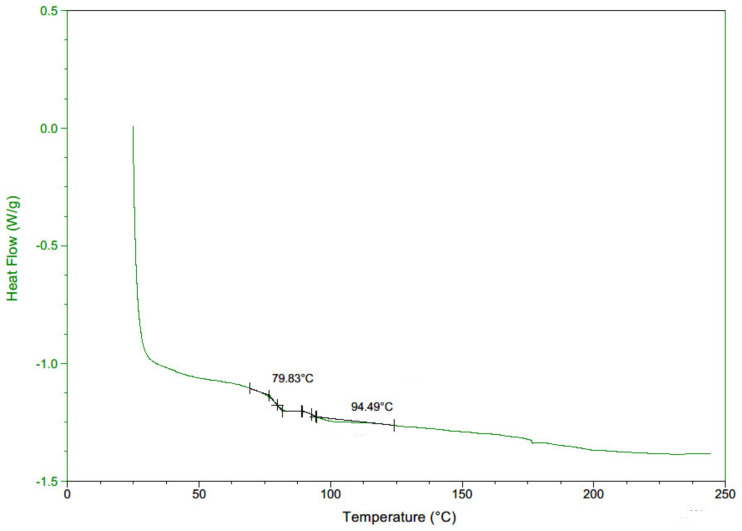
DSC of Ultimaker red ABS.

**Figure 8 polymers-14-00982-f008:**
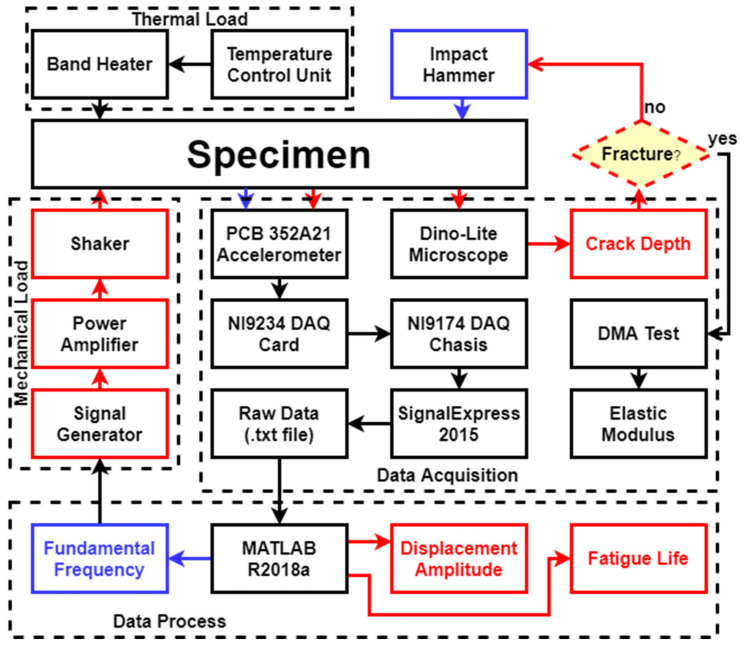
Experimental procedures.

**Figure 9 polymers-14-00982-f009:**
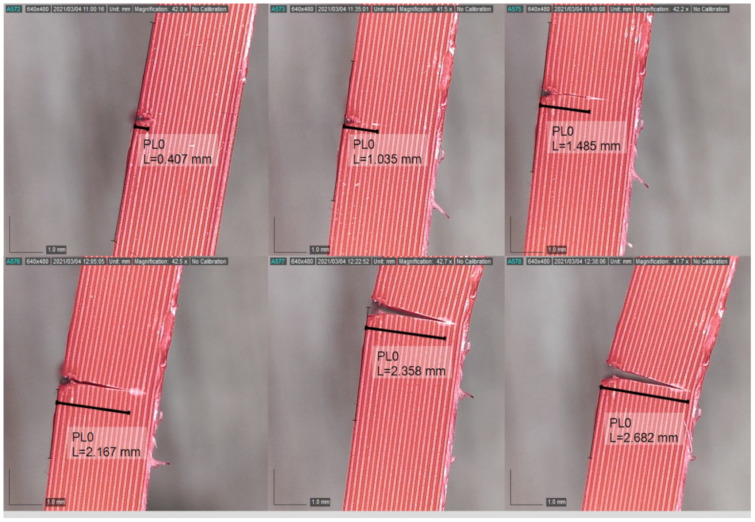
Crack growth process at 10 mm crack location from initial seeded crack (0.407 mm) to final crack (2.682 mm) at 70 °C.

**Figure 10 polymers-14-00982-f010:**
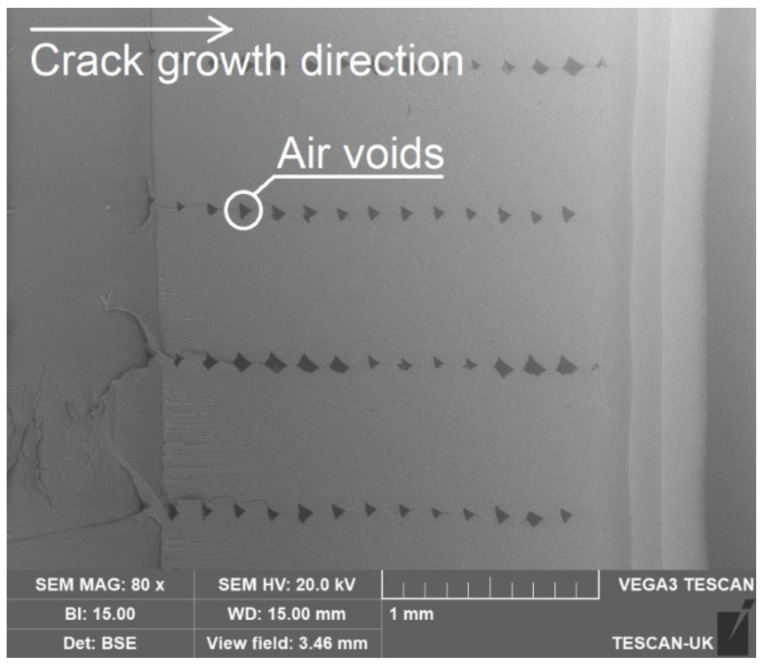
Fracture surface of the specimen after fracture.

**Figure 11 polymers-14-00982-f011:**
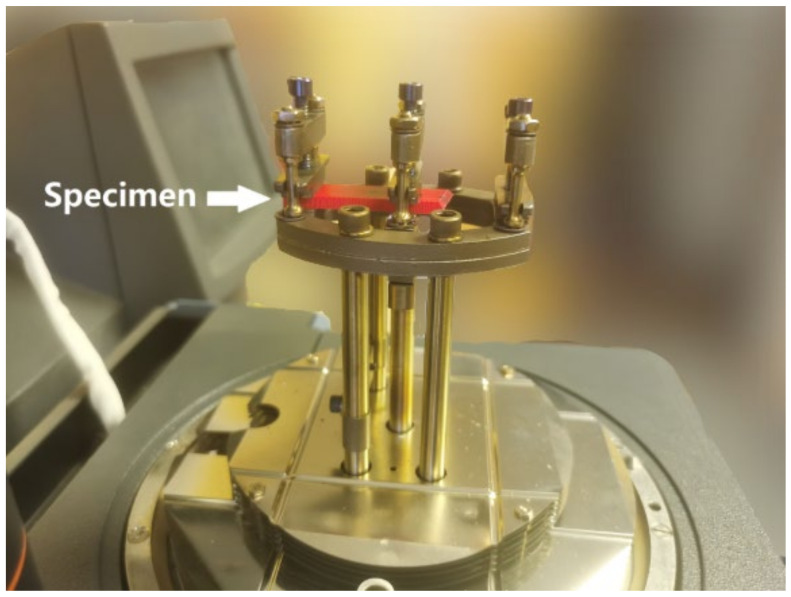
Single-clamped DMA test setup.

**Figure 12 polymers-14-00982-f012:**
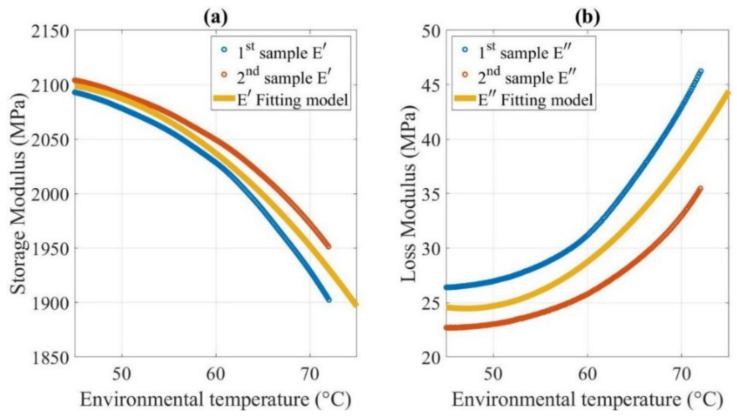
(**a**) Storage modulus and (**b**) loss modulus of FDM ABS.

**Figure 13 polymers-14-00982-f013:**
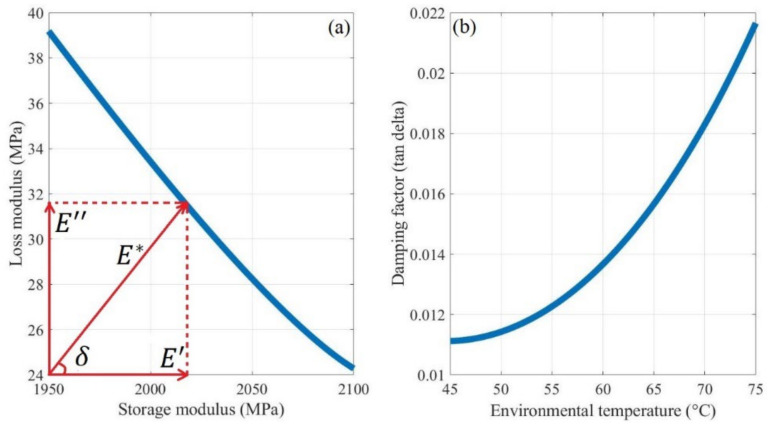
(**a**) Visualized relationship among the storage modulus, loss modulus, and damping factor. (**b**) Change in the damping factor with temperature.

**Figure 14 polymers-14-00982-f014:**
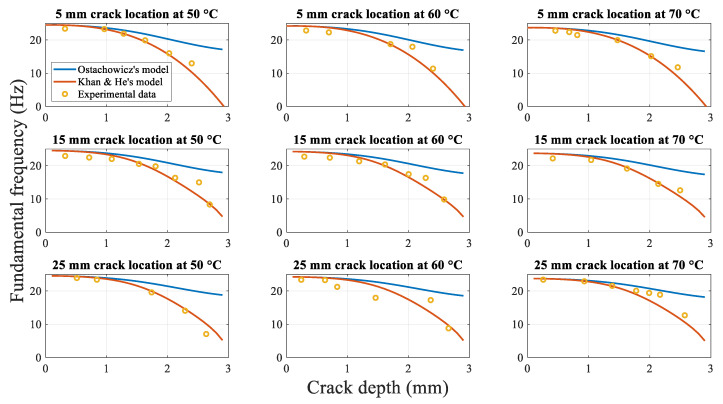
Fundamental frequencies obtained experimentally and calculated using two analytical models during crack propagation.

**Figure 15 polymers-14-00982-f015:**
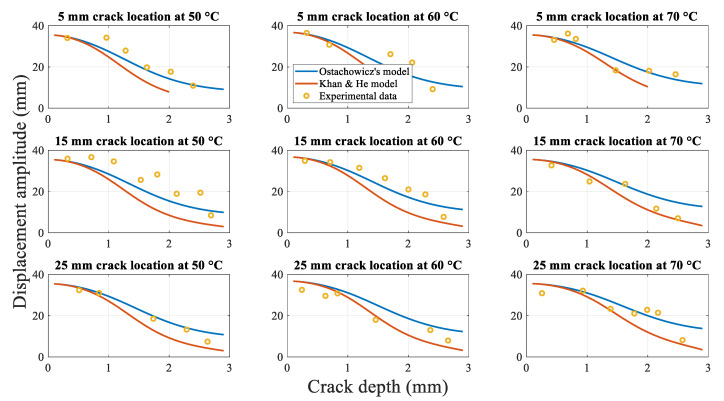
Analytical displacement amplitudes obtained experimentally and calculated using two analytical models during crack propagation.

**Figure 16 polymers-14-00982-f016:**
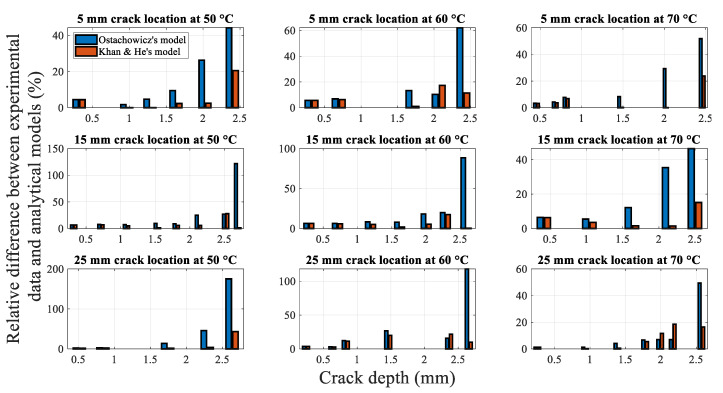
Relative difference in the results obtained using analytical models and experimental results.

**Figure 17 polymers-14-00982-f017:**
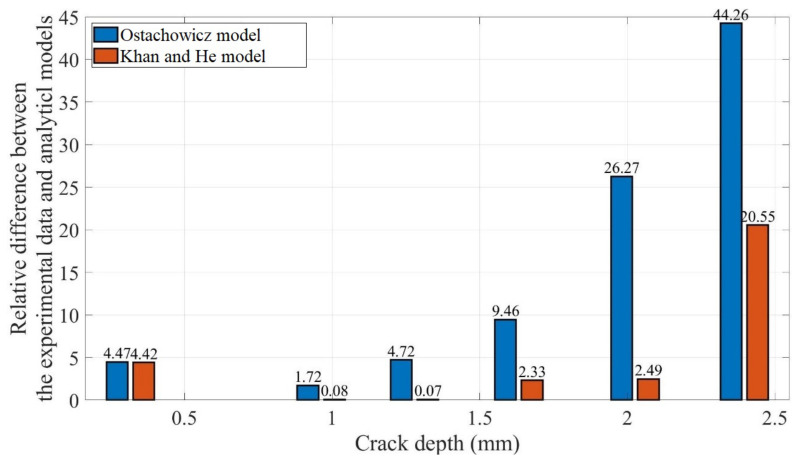
Relative difference between the fundamental frequency obtained using the two analytical models and experimental data for crack propagation at the 5 mm crack location at 50 °C.

**Figure 18 polymers-14-00982-f018:**
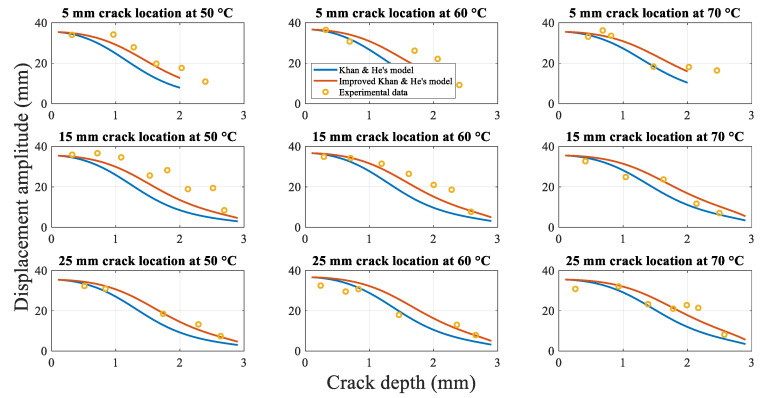
Displacement amplitudes obtained using the analytical methods and experiment during crack propagation.

**Figure 19 polymers-14-00982-f019:**
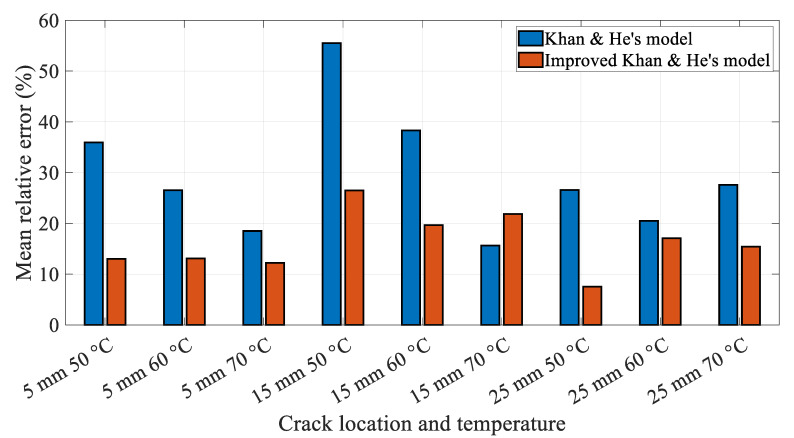
Mean relative difference for different combinations of the crack locations and temperatures.

**Figure 20 polymers-14-00982-f020:**
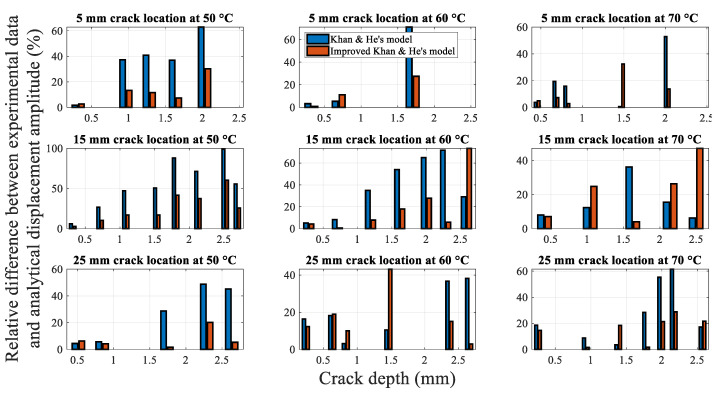
Relative difference between models and experimental data during crack propagation.

**Figure 21 polymers-14-00982-f021:**
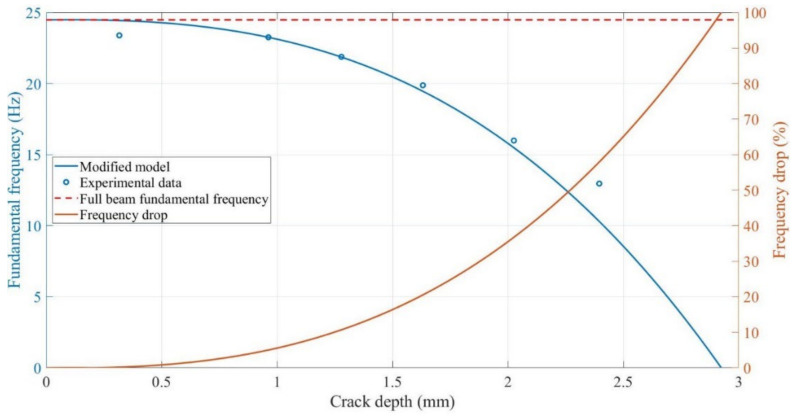
Change in frequency during crack propagation at the 5 mm crack location at 50 °C.

**Figure 22 polymers-14-00982-f022:**
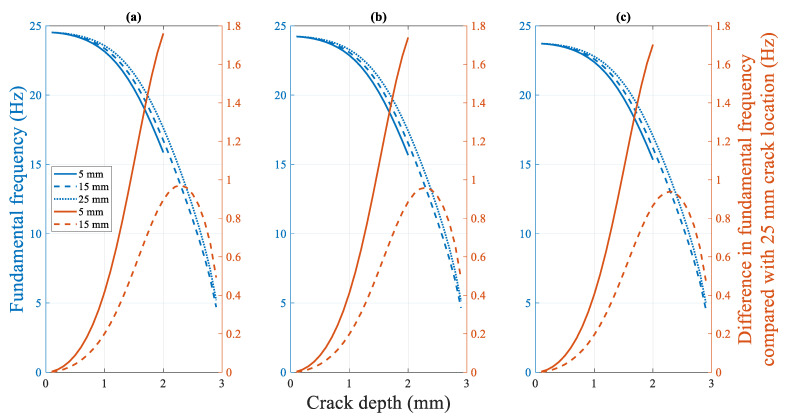
Difference in the fundamental frequency with crack locations: Crack propagation at (**a**) 50 °C (**b**) 60 °C (**c**) 70 °C.

**Figure 23 polymers-14-00982-f023:**
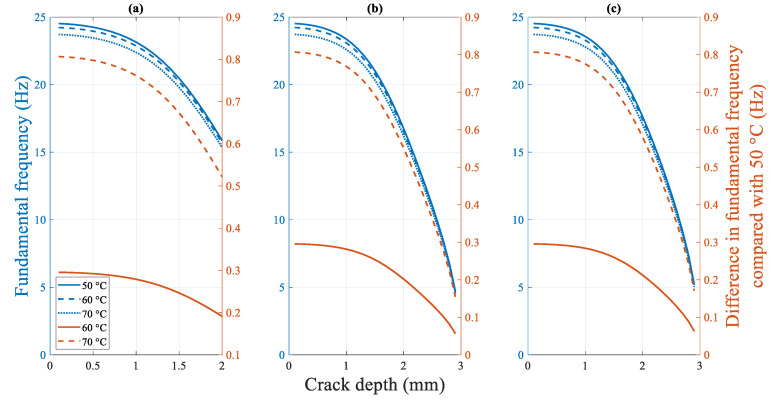
Difference in the fundamental frequency of the beams under different temperatures at a crack location of (**a**) 5 mm (**b**) 15 mm (**c**) 25 mm.

**Figure 24 polymers-14-00982-f024:**
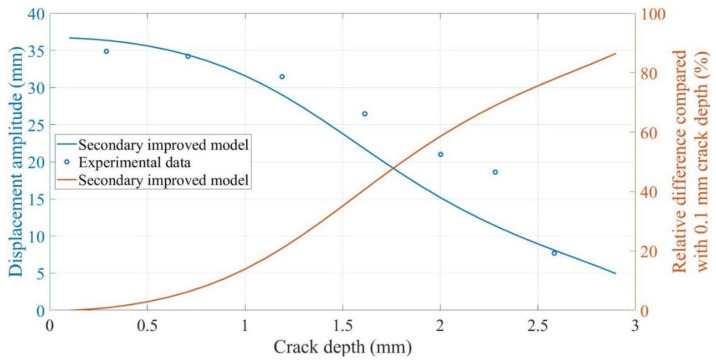
Change in the displacement amplitude for crack growth at a 15 mm crack location at 60 °C.

**Figure 25 polymers-14-00982-f025:**
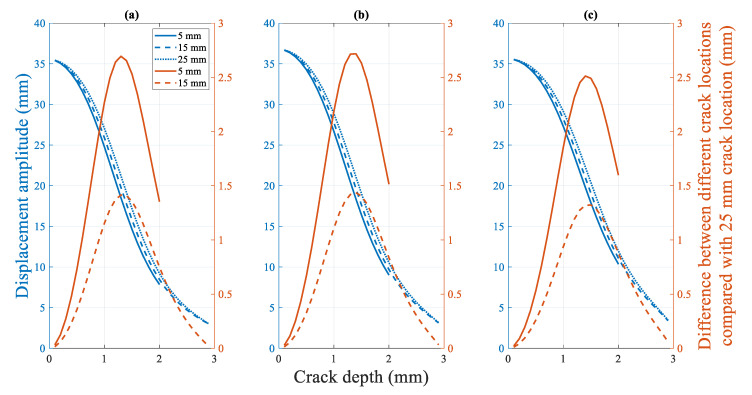
Difference in the displacement amplitude for a beam with different crack locations, with crack propagation at (**a**) 50 °C (**b**) 60 °C (**c**) 70 °C.

**Figure 26 polymers-14-00982-f026:**
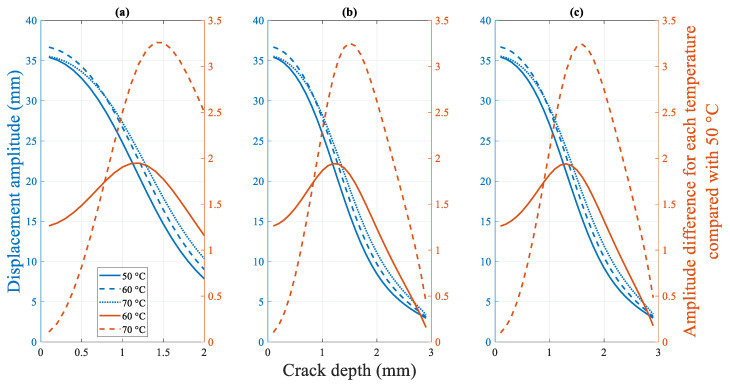
Difference in the displacement amplitude for the cracked beam under different temperatures at crack locations of (**a**) 5 mm (**b**) 15 mm (**c**) 25 mm.

**Figure 27 polymers-14-00982-f027:**
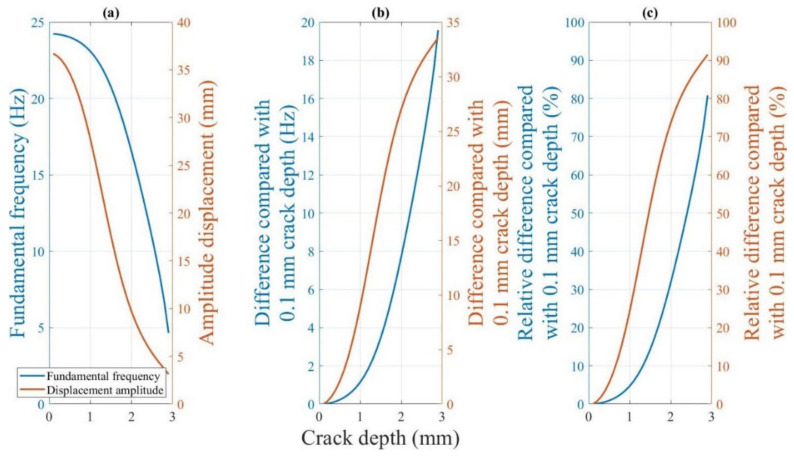
(**a**) Dynamic response of the beam during crack growth at a 15 mm crack location at 60 °C (**b**) Difference in the dynamic response with that for a 0.1 mm crack depth. (**c**) Percentage difference in the dynamic response with that for a 0.1 mm crack depth.

**Figure 28 polymers-14-00982-f028:**
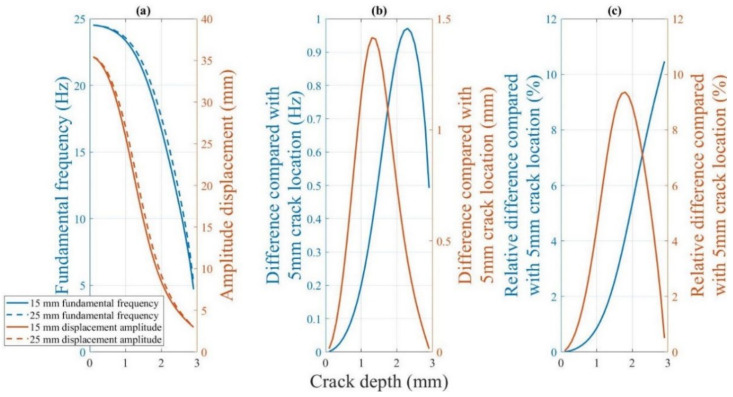
(**a**) Difference in the dynamic responses of the beam with 5 and 15 mm crack locations at 60 °C. (**b**) The difference in the dynamic responses. (**c**) The relative difference in the dynamic response corresponding.

**Table 1 polymers-14-00982-t001:** Main printing parameters for the FDM ABS cantilever beam.

Printing Parameters	Settings
Infill density	100%
Raster orientation	0°
Nozzle size	0.8 mm
Layer thickness	0.15 mm
Nozzle temperature	245 °C
Bed temperature	90 °C

**Table 2 polymers-14-00982-t002:** Fundamental frequency obtained from experiments and analytical models, along with the difference in the values.

Crack Depth (mm)	0.317	0.963	1.279	1.632	2.026	2.396
Experimentally obtained fundamental frequency $f_{test}$ (Hz)	23.39	23.26	21.88	19.88	15.99	12.96
Fundamental frequency $f_{1}$ (Hz) obtained using Ostachowicz model	24.44	23.66	22.91	21.76	20.19	18.7
Fundamental frequency $f_{2}$ (Hz) obtained using Khan-He model	24.42	23.27	21.9	19.42	15.59	10.3
Difference between $f_{test}$ and $f_{1}$ (Hz)	1.04	0.4	1.03	1.88	4.2	5.74
Difference between $f_{test}$ and $f_{2}$ (Hz)	1.03	0.018	0.014	0.46	0.4	2.66
Difference between $f_{test}$ and $f_{1}$ (%)	4.47	1.72	4.72	9.46	26.27	44.26
Difference between $f_{test}$ and $f_{2}$ (%)	4.42	0.08	0.07	2.33	2.49	20.55

## Data Availability

The data presented in this study are available on request from the corresponding author.

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
