# Peer review of "Interdependencies between Dynamic Response and Crack Growth in a 3D-Printed Acrylonitrile Butadiene Styrene (ABS) Cantilever Beam under Thermo-Mechanical Loads"

_polymers, 2022, doi:10.3390/polym14050982_

Round 1
Reviewer 1 Report
The article presents a unique and significant topic. The results obtained and discussed were outstanding, and they can be relied upon to improve 3D printing.
It would have been preferable for the authors also to use the SEM test to check the shape and size of the crack.
They can also use a DSC test not just DMA.
What about rheological properties?
Reviewer 2 Report
This paper proposes an analytical model to determine the dynamic response of a cracked 3D-printed ABS cantilever beam subjected to a thermo-mechanical load. Compared with the existing model, the fundamental frequency can be modelled more precisely. The corresponding displacement amplitudes were calculated with the consideration of the crack breathing phenomenon. The experimental results validated the proposed model. The article is concise and very well written. And it fits the theme of the journal. I recommend acceptance of the paper after minor revisions.
- Remove the bullet points on Page 6 and describe the boundary conditions in a paragraph, consistent with the style of a journal article.
- In figures 3-1 and 3-3, the materials in the background should be removed. They have nothing to do with the research.
- Typos and grammar errors are throughout the article, and they should be removed. For example, the first sentence “fused deposited modelling (FDM) is used extensively…” should be changed to “Fused deposited modelling (FDM) is used extensively…” Always write out the first in-text reference to an acronym, followed by the acronym itself written in capital letters and enclosed by parentheses, such as NASA.
Round 2
Reviewer 1 Report
Unfortunately, the author has not adhered to the reviewing notes. His answers were not convincing. The information you requested is essential for a better understanding of the mechanism. He says in all his answers that he agrees with the question and the demand, but at the same time, he did not implement it or put an alternative to it.
I recommend not to accept the manuscript until making the required modifications, as they are very necessary.
Round 3
Reviewer 1 Report
I recommend accepting the article now